# Mentally Demanding Work and Strain: Effects of Study Duration on Fatigue, Vigor, and Distress in Undergraduate Medical Students

**DOI:** 10.3390/healthcare11121674

**Published:** 2023-06-07

**Authors:** Gerhard Blasche, Tav A. K. Khanaqa, Michaela Wagner-Menghin

**Affiliations:** 1Center for Public Health, Department of Environmental Health, Medical University of Vienna, 1090 Vienna, Austria; 2Department of Psychiatry and Psychotherapy, Clinical Division for Social Psychiatry, Medical University of Vienna, 1090 Vienna, Austria

**Keywords:** academic stress, work hours, mental work, studying, fatigue

## Abstract

Aims: The impact of the extent of mentally demanding work on the next-day’s strain is largely unknown, as existing studies generally investigate consequences of extended versus normal workdays. The present study sought to fill this gap by investigating how short work periods of mentally demanding academic work impact strain reactions in medical students preparing for an exam, using days of no work as reference category. Method: The observational design involved students repeatedly self-reporting fatigue, vigor, distress, and the preceding day’s study duration. Hours of nocturnal sleep, attending paid work and compulsory classes, gender, and proximity to the exam were controls in the linear model (generalized estimating equations). Forty-nine students provided 411 self-reports (M = 8.6, SD = 7.0 self-reports/student). Results: Engaging in mentally demanding work was associated with increased distress and work periods > 4 h with increased fatigue. Distress, vigor loss, and fatigue increased in proximity to the exam. Conclusion: Despite students’ high control of their schedule, even short periods of mentally highly demanding work may impair next-day’s well-being when task motivation is high. Freelancers and students might require health-promoting scheduling of work and leisure to avoid an accumulation of strain.

## 1. Introduction

The impact of mentally demanding work on strain reactions, including fatigue and distress, is of imminent relevance for the performance, safety, well-being, and health of those working and is the basis for working time legislation, working time arrangements, and the individual’s planning of work and leisure. In general, mentally demanding work causes strain reactions that increase in magnitude with the duration of work [1]. This agreement rests on a large body of empirical evidence from different study designs and populations. Experimental studies (e.g., [2]) commonly include volunteers and observe the increase in strain reactions throughout short periods of demanding tasks (e.g., their duration usually expressed in minutes). Observational (e.g., [3]) as well as cross-sectional (e.g., [4]) field studies typically include employees (e.g., [3]) or, less often, self-employed persons (e.g., [5]) or students (e.g., [6]) and observe strain reactions following longer work-periods. Results of experimental studies show short term increases in fatigue and a worsening of performance. Results of field studies show that working more than 50 [4] or 60 h [7] per week or working more than 8 h per day [8,9] is associated with higher levels of fatigue compared to working 50 or 60 h per week or less, respectively, or 8 h per day or less. 

The Effort–Recovery Model [10] maintains that workload is associated with strain reactions because subjects’ efforts to perform deplete physiological and psychological resources. This depletion is reversible, given the opportunity for full recovery. However, it should be pointed out that there are also other models of mental fatigue such as the opportunity cost model, which views fatigue as consequence of motivational processes [11,12]. In addition, it has been shown that frequent changes between high demands and recovery contribute to the building up of physiological toughness [13], whereas incomplete recovery contributes to the accumulation of strain and impacts health and well-being negatively [14]. This suggests that healthy work and break schedules, especially for mentally demanding work, have the ability to foster resilience while preventing the accumulation of strain (see, e.g., [15]). 

Students typically are subject to high levels of demand. Rates of depression are higher amongst university students as compared to the general population [16], and level of distress are especially high amongst medical students [17]. Students having concerns about their marks have a higher risk of experiencing stress and depression compared to students with fewer concerns [18]. Similar effects also have been found for nursing students [19]. For undergraduate medical students, a recent study indicated a decrease in psychological well-being in the first term of their curriculum due to mental overload and performance pressure [20]. Thus, in order to prevent a depletion of psychological and physiological resources, healthy academic work schedules need to be developed. As stated before, these schedules must adequately space demands and recovery periods to prevent adverse effects while, at the same time, increasing resilience. 

Although there is a large body of empirical evidence regarding work durations’ effects on strain reactions, it is challenging to derive recommendations for an optimal scheduling of mentally demanding work from this evidence. First, studies markedly differ in observed duration and scheduling of work and/or task demands. While experimental studies typically expose subjects to tasks of several minutes [2,12] to several hours [21,22] without the inclusion of rest breaks, field studies involving employees typically observe the effects of work days of eight or more hours [8,9] or total work-weeks [4,7], thereby including usual break and leisure times. Thus, evidence for intermediate durations of demanding work on strain reactions is scarce. Second, there are differences in participants’ task control and motivation, both of which affect the increase in fatigue. According to the compensatory control model, control is essential for curbing the increase in strain [23,24]. According to the motivation intensity theory, high personal relevance of a task leads to a high investment of effort, which, in turn, increases fatigue [25,26]. Volunteers in experimental studies and employees in field studies might have less control over the scheduling of tasks compared to self-employed persons or students. However, individuals engaged in paid work and students might also show higher motivation to accomplish a task, due to the potentially more severe negative consequences of poor performance. 

The present study sought to fill this gap by investigating how short periods of mentally demanding work in high-control and high-motivation settings impacted strain reactions in students preparing for an exam. Studying for an exam can be considered an ecologically valid type of mentally demanding work. Indeed, studying is associated with higher levels of perceived stress and cortisol, both of which increase with study duration and/or perceived academic demands [6,27,28]. One can expect students preparing for an exam to have high levels of goal commitment (i.e., motivation to perform well) as well as greater control of their working time compared to both individuals participating in experimental studies as well as working individuals in field studies. Thereby, the present study contributes information needed to plan healthy work and break schedules for settings with mentally demanding work characterizes by high control and high motivation, such as self-employment or studying.

We chose to study the effects of study duration on three variables typically used to assess subjective strain reactions: fatigue, vigor, and distress. Whereas fatigue reflects the experience of weariness and decreased capacity for work following mental or physical exertion [29], vigor is a positive mental state reflecting the willingness and ability to engage in tasks [30]. To fully capture the effects of demanding work, we included distress, which increases with demands, according to the psychological theory of multidimensional activation [30]. Following the Effort–Recovery Model [10], we expect study duration to be positively associated with fatigue (hypothesis one) and distress (hypothesis two) and negatively with vigor (hypothesis three) after controlling among others for sleep duration (e.g., [31]) and exam-induced stress (e.g., [32,33]).

## 2. Materials and Methods

### 2.1. Study Participants

Study participants were medical students in their second semester at the Medical University of Vienna/Austria, a Central-European public university. The full-time six-year medical program is taught in German Language without tuition fee. Students with high school degrees who master the entry test, which includes a complex array of cognitive performance tests, are enrolled. Being in their second semester, the students included in the study already had time to settle in. They will have written some course exams as well as their first end-of-term exam, which includes all topics of the first semester. Thus, they will have gained a good overview of their schedule and the effort required to thrive in their academic work. Thus, we regard this student population as highly motivated and highly skilled regarding academic work while having adequate control in the scheduling of their study time. 

Students were recruited by approaching them in their compulsory classes and inviting them to participate in a study on the effect of academic stress on well-being. Fifty-nine participants gave informed consent; forty-nine students met the final inclusion criteria (see point 2.5 data selection). The mean age was 20.6 (SD 2.3) years, varying between 18–29 years (three did not disclose their age). Thirty-one participants were female (63%), thirteen male (26%), and five (10%) did not disclose their gender. Nineteen participants (43%) engaged in employed work, with a maximum of 20 h per week.

### 2.2. Design

Daily fatigue, vigor, and distress were assessed repeatedly during three phases of the semester to account for variations in academic workload throughout the term. 

Phase 1: Five consecutive days (Mon–Fri, in March) ten weeks before the final exam constituted phase 1. During this week, students attended compulsory classes for a total of 24 h.Phase 2: The Tuesdays in the four weeks before the exam (the last of these Tuesdays being the day before the exam, in June) and the exam day constituted phase 2.Phase 3 included the eleven days (Thur–Sun) following the exam in July. During phases 2 and 3, no compulsory classes took place.

Participants received an invitation and further instructions from T.K. on the first day of phase 1, the four Tuesdays of phase 2, and the first day of phase 3. 

Typically, diurnal sleepiness decreased until around midday and remained constant until early afternoon, after which it increased again until bedtime [34,35]. As such, the level of fatigue at a given time point equaled the sum of task-related fatigue and diurnal sleepiness [36], making the contribution of work duration to the increase in fatigue unclear. To control for these circadian effects, study participants reported their strain reactions as close to 18:00 as possible. To account for participants’ scheduling difficulties and individual circadian preferences, data entries between 14:00–22:00 were accepted.

Students’ data were recorded using an online tool (www.soscisurvey.de (accessed on 10 July 2015)) configured to allow only the submission of complete daily self-reports.

### 2.3. Variables

#### 2.3.1. Independent Variable—Study Duration

Study duration was based on students’ self-reporting of the previous day’s hours spent studying. Participants responded to the question “How many hours did you study yesterday?” and provided the number of hours using a pull-down menu providing 0 to 12 h in full-hour steps. This hourly categorization was chosen initially to provide later results on strain development in hourly steps. However, to avoid small cell numbers in the analysis, data had to be collapsed into five categories (0 h, 1–2 h, 3–4 h, 5–6 h, 7 or more hours).

#### 2.3.2. Control Variables

Sleep duration of the preceding night (self-reported) was included to control for present-day sleepiness.Proximity to exam (i.e., the number of days preceding or following the exam, based on the time stamp in the online tool) was included to control for variations in students’ changes in examination stress throughout the term.The previous day’s compulsory classes (yes/no, based on class schedule) and the previous day’s paid work (yes/no, self-report) were included to control for changes in the ability to control one’s study schedule because of other responsibilities.Time of questionnaire completion (based on the time stamp in the online tool) was included to control for strain-related effects of the circadian rhythm.Additionally, gender (self-report) was included.

#### 2.3.3. Dependent Variables

The dependent variables subjective fatigue, vigor, and distress were chosen, as these variables are typically included in studies on subjective strain reactions. They were assessed with the Personal State Scale, a validated and widely used German questionnaire assessing various facets of mental strain [37]. Participants responded to the question “Please indicate how you feel in the present moment” using adjectives to be rated on a 6-point rating scale (1 = “hardly”; 2 = “a bit”; 3 = “somewhat”; 4 = “rather”; 5 = “predominantly”; and 6 = “completely”). 

Fatigue was assessed with eight adjectives (e.g., “tired”) and had a scale reliability (Cronbach’s Alpha) of CA = 0.91.Vigor was assessed with seven adjectives (e.g., “full of energy”) with a scale reliability of CA = 0.94.Distress was assessed with five adjectives (e.g., “nervous”) and had a scale reliability of CA = 0.85.

### 2.4. Data Analysis

#### 2.4.1. Descriptive Statistics

Descriptive statistics on independent and control variables for the three study phases were computed to evaluate the assumption of variations in students’ workloads and abilities to control their schedule.

#### 2.4.2. Hypotheses Testing

A linear model (Generalized Estimating Equations, GEE, SPSS 26) was used to test the three hypotheses. GEE is a regression-based approach to handle clustered data, such as families [38] or longitudinal and repeated measurement data [39], and can be used with binary and categorical data. As such, we consider it ideal for modeling study duration effects on strain reactions in a field study setting. Calculations were based on a linear model using a robust estimator adjusted by the number of non-redundant parameters and an independent working correlation matrix. A subject variable was entered to account for repeated measures. Wald Chi-square statistics were used to assess the hypotheses and the effect of the control variables. Regression coefficients were used to illustrate the magnitude of the effect. The SPSS/GENLIN algorithm was used to estimate model parameters and perform statistical testing. However, GENLIN currently does not provide effect sizes. The dependent variables fatigue, vigor, and distress were entered into separate analyses. The independent variable was study duration. As control variables, sleep duration (continuous), time of questionnaire completion (continuous), proximity to exam (continuous), days with compulsory classes (categorical), days with paid work (categorical), and gender (categorical) were included.

#### 2.4.3. Transformation

The variable “proximity to exam” was built to account for the non-linear association between the proximity to the exam and examination stress [33]. Stress was found to dissipate disproportionately relative to the proximity to the exam. Thus, the number of days preceding or following the exam were transformed using the power function ((x + 1)^(−1)^) with (−1) as an exponent. A pretest resulted in a better model fit than a linear function or a power function with (−2) as an exponent (results not shown). According to exploratory data analysis (results not shown), distress did not peak on the day of the exam, as fatigue and vigor did, but did one day before the exam (keeping in mind that strain was assessed after the exam). Thus, an additional linearly transformed temporal distance variable was calculated to account for the distribution for distress.

### 2.5. Data Selection

From the consenting 59 students, 10 had to be excluded: One individual was not included due to employed work of more than 20 h per week, which was against the recommendation for studying in the full-time program.The data from three students were excluded because the time stamp revealed that they self-rated their daily subjective strain “on bulk” for several days at the end of each phase instead of completing the questionnaire each day, as instructed.The data of six students were excluded because they were not compliant with the instruction to complete the rating as close to 18:00 as possible. Completing the rating after 22:00 or before 14:00 was arbitrarily defined as non-compliant.

The final data set included 411 observations provided by 49 individuals. The average number of observations per person was 8.6 (SD 7.0, range 1–21), indicating low compliance with the study, as most participants provided data on less than half of the defined observation days.

## 3. Results

### 3.1. Descriptive Statistics

Descriptive statistics of the independent and control variables are provided in total as well as for the three phases of the academic semester (Table 1). Consistent with our assumption of varying workload across the three phases, study duration was longest within the four weeks before the exam, shortest post-exam, and intermediate in the week located 10-weeks before the exam, (Waldχ^2^ (2, *N* = 411) = 69.554, *p* < 0.001). 

Of the control variables, average sleep duration was 7.4 h, sleep being the longest post-exam (Waldχ^2^ (2, *N* = 411) = 21.695, *p* < 0.001). The variable proximity to exam was built to control for examination stress, assuming examination stress to be highest in the days before and after the exam. Students participated in compulsory classes only during phase one. Less than 10% of students reported having been busy with paid work on the days they reported to have been studying, with no differences between phases (Waldχ^2^ (2, *N* = 411) = 4.272, *p* = 0.118). We included the time of questionnaire completion to control for circadian influences and compliance with instruction to complete the self-rating as close to 18:00 as possible. In accordance with instructions, the average questionnaire completion time was 18.4 h, though it was slightly later during phase one than during the other two phases (Waldχ^2^ (2, N = 411) = 8.385, *p* < 0.015). Table 1 provides descriptive statistics.

### 3.2. How Do Short Work Periods of Mentally Highly Demanding Work Impact Strain Reactions in Students Preparing for an Exam?

GEE analysis was used to test if study duration significantly predicted an increase in strain reactions while controlling for the influence of diurnal sleepiness, circadian rhythm, exam-related changes in stress, and control of schedule on strain reactions. The independent variables fatigue, vigor, and distress assessing strain reactions were entered in three independent analyses.

#### 3.2.1. Fatigue (Hypothesis One)

In general, *fatigue* was found to increase with previous day’s study duration (Waldχ^2^ (4, *N* = 411) = 26.3, *p* < 0.001), thus supporting hypothesis one. In particular, studying 5–6 h (Waldχ^2^ (1, *N* = 411) = 6.923, *p* = 0.009) or more (Waldχ^2^ (1, *N* = 411) = 23.023, *p* < 0.001) on the previous day was associated with an increase in the present day’s fatigue (Figure 1, blue line).

Additionally, fatigue increased with increasing proximity to the exam and decreasing sleep duration. Attending compulsory classes, engaging in paid work, the time of questionnaire completion, and gender were not associated with fatigue. See Table 2 (column fatigue) for statistics.

#### 3.2.2. Vigor (Hypothesis Two)

Vigor was not associated with study duration (Waldχ^2^ (4, *N* = 411) = 2.124, *p* = 0.713); thus, hypothesis two, predicting a study duration-related loss of vigor, was not supported. (Figure 1, red line). As displayed in Table 2 (column vigor), vigor was also not related to any of the control variables of the model, except for proximity to the exam, indicating reduced vigor immediately before and after the exam.

#### 3.2.3. Distress (Hypothesis Three)

Distress was found to increase with the previous day’s study duration (Waldχ^2^ (4, *N* = 411) = 44.2, *p* < 0.001), thus supporting hypothesis three (Figure 1, green line). Studying 1–2 h or more yesterday as compared to not studying went along with higher distress in the present day. Additionally, distress was not related to any other model variables, except for proximity to exam indicating higher distress before and/or after the exam (all statistics provided in Table 2, column distress).

## 4. Discussion

Medical students, who are highly motivated to study and have a high degree of control over their schedule, feel increasingly distressed and experience increasing fatigue the more they engage in mentally demanding work. Even having studied for as little as one or two hours on the previous day was associated with increased distress in the present day. Fatigue increased after studying for more than 4 h on the previous day compared to not studying at all. Vigor, however, was unrelated to the previous day’s study time, but students experienced reduced vigor immediately before and after the exam. 

The relationship between study duration and fatigue was in accordance with existing studies on work duration and fatigue in paid work [4,7,8,9]. The present study on academic work in students added to the existing literature by investigating the effects of shorter work periods (<8 h) and using a more fine-grained analysis. As opposed to studies on paid work, which typically provide results on workdays exceeding six hours of work or more using an 8 h workday as a reference category [8,9] or a work week of 50 h or more using a 40 h week as a reference category [4,7], we used days with no work at all as reference category. As such, we could show that academic work of up to four hours was not associated with increased fatigue compared to no work, whereas work exceeding four hours increased fatigue. However, academic work may be mentally more demanding than the tasks required during paid work [33,40,41], therefore possibly contributing to fatigue to a greater extent. 

Study duration was also positively associated with distress, which increased roughly linearly with study duration. This association was apparent, despite controlling for the proximity to the exam and for class attendance, indicating that the previous day’s workload led to increased levels of tense arousal. A close association between the extent of academic workload in hours and perceived stress in students was found in a recent study [28], thus confirming our results. Associations between working time and perceived stress or anxiety also have been found for paid work both in longitudinal [42,43] as well as in cross-sectional studies [44,45], indicating that long hours of mentally demanding work not only increase fatigue but also lead to higher levels of distress. Indeed, a reciprocal association between fatigue and distress has been found in students, indicating that distress not only causes fatigue, but fatigue also causes distress [46]. 

Contrary to our expectations and despite the associations with subjective fatigue and distress, study duration was not associated with vigor. When defining vigor as the ‘opposite of fatigue‘, one would expect a decrease in vigor with working time. Indeed, this has been found in experimental studies subjecting individuals to demanding mental tasks [22]. However, as opposed to negative emotional states such as fatigue or burnout, positive emotional states such as vigor or engagement are associated with work demands that are perceived as challenging in real-life work. This is the case because “challenging situations promote engagement when employees trust their investment of time and energy will be rewarded in some meaningful way” [47]. Indeed, a positive association between challenge demands and engagement has been found in several meta-analyses [48,49]. Even though our study did not find that study hours were positively related to vigor, the failure to find an association between study hours and vigor suggested that students preparing for a course or an exam invested more effort when the workload was high, thereby counteracting increases in fatigue. This finding is aligned with the motivational control theory, which proposes that “at least for highly motivated tasks, primary task goals appear to be stabilized by increased regulatory control, with associated costs” [50]. 

In general, our results indicated that the strain reaction associated with five of more hours of mentally demanding work—as compared to days without any work—spilled over to the next day, thereby affecting next-day’s fatigue. Thus, five or more hours of mentally demanding work may exceed the individual’s ability for full recovery, thereby allowing for a build-up of fatigue from one day to the next. Whereas such a build-up of fatigue was usually not observed for the normal 8 h workday in employees [51], it is found for 12 h shifts, where the next-day’s fatigue is higher than on the previous day [52]. The present results suggested that this build-up could already occur for work durations exceeding 4 h if the work is sufficiently demanding. 

A note on the effect of the control variables seems warranted. All three strain variables were associated with the proximity to the exam, reflecting examination stress [53]. As this variable was based on an inverse power function and was thus non-linear, this association predominantly reflected strain on the exam day and the days immediately preceding and following the exam. Indeed, in a previous study, recovery from academic exam-related strain reactions took up to a week for fatigue and several days for distress [33]. In addition, fatigue in the present study was also associated with sleep duration, a finding also being in line with literature [54,55]. It should be noted that attending compulsory classes was not associated with next-day strain, despite the known detrimental effects of lectures on vigilance [56,57]. This suggested that these adverse effects of class attendance are of short duration. 

The strengths of our study were threefold. First, our observational field study setting made it possible to observe subjects engaging in the mentally demanding work of learning for their end-of-year exam in medical school, a highly relevant task for first-year students. Second, as observations were conducted repeatedly throughout the spring term, they covered a typical student work cycle characterized by varying work hours depending on task motivation and academic requirements. It was thus possible to observe the effects of work hours on strain reactions and related variables in a within-person design with days of no work as reference category. Third, strain assessments were made at a specified time span during the day, thus limiting circadian effects. 

### Limitations

A limitation of our study was that we did not control for if and how long subjects engaged in recovering activities or in mentally demanding work on the previous or present day, respectively, despite the study investigating the previous day’s study duration on present day’s fatigue. Although it is reasonable to assume that study durations on adjacent days were similar, the precise effect of the same day’s study duration remained unclear. Time spent on recovery activities during the day is strongly related to daily well-being [58] and fatigue [59]. The possibility of recovery in our study may explain the discrepancy of our results to those of experimental studies, the latter finding increases in fatigue essentially after the first few minutes of the time on the task. However, experimental studies generally do not provide opportunities for recovery, whereas we can assume that participants in our study did take some time off [22,60]. Additionally, the previous day’s study duration was assessed retrospectively, potentially contributing to inaccuracies due to recall bias [61].

Another limiting aspect was that the relationship between activation and behavioral efficiency and affect resembled an inverted U according to the activation theory [62]. This may have systematically distorted the assessment of subjective strain, assuming that work stimulates individuals to a greater extent than the non-work activities. Indeed, a non-linear relationship between workload and fatigue has been found in several studies, indicating that fatigue initially decreases when work becomes more engaging but increases again after a certain level of workload [63,64]. Thus, it seems likely that measures of subjective fatigue during work such as studying for an exam may underestimate fatigue due to work-related activation. Future studies would need to additionally assess present-day study and recovery activities and/or implement a design allowing the observation of subjects’ activities more closely, possibly using multiple assessments throughout the day. 

## 5. Conclusions

In general, for paid daytime work, it is well established that strain reactions increase relevantly when work duration exceeds the standard working time of 6 or 8 h/day or 40 h/week. For students’ academic work, we found that shorter studying periods contribute to next-day fatigue and distress, compared to days with no academic work. Still, despite the varying length of daily work periods throughout the spring term, student vigor was unrelated to study duration. 

Theoretically, our study elaborates on the complex relationship between the effort to perform, the opportunity for recovery, and physiological and psychological strain reactions, as illustrated by the Effort–Recovery Model [10]. However, straining part of one’s physiological and psychological resources for short periods (hours) and experiencing fatigue and distress for some weeks does not lead to loss of vigor, at least when the task is motivating and highly relevant and when scheduling of work and recovery time is possible for the individual. Nevertheless, our results tentatively suggest that working for more than four hours may overtax the individual’s possibility of daily recovery, leading to a build-up of fatigue from one day to the next.

On a practical level, our results are relevant for persons working on projects requiring mentally demanding work spaced over some weeks until a deadline, who also have high control over their schedule, such as self-employed persons or students. Under these conditions, it is prudent to counteract the buildup of fatigue and distress occurring after five hours of work or less by planning sufficient recovery times on a daily basis and/or limiting the hours of mentally demanding work to a maximum of four hours per day to avoid a buildup of fatigue. If this is not possible, a sufficient number of free days should be scheduled after the phase of high demands to permit full recovery [33]. 

Suggestions for future studies are to determine the daily extent of mentally demanding work leading to increased next-day’s strain reactions while, at the same time, accounting for recovery. In addition, strain should be assessed in the morning in advance of daily work demands. Furthermore, healthy daily and weekly ratios of mentally demanding or less-demanding work, recovery times, as well as sleep should be determined by the use of multiple daily assessments over a time span of several days to weeks. 

## Figures and Tables

**Figure 1 healthcare-11-01674-f001:**
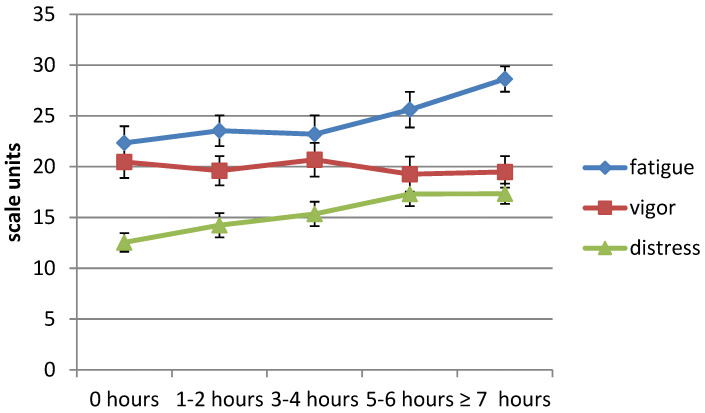
The association of yesterday’s study duration with strain; observed adjusted means and standard errors.

**Table 1 healthcare-11-01674-t001:** Descriptive statistics (unadjusted for correlation, mean (m) and standard deviation (sd) or frequencies (n) and percentages (%)) for independent variables and control variables.

	Phase 1 10 Weeks Pre-Exam 5 Days		Phase 2 4–1 Week(s) Pre-Exam 5 Days		Phase 3 Post-Exam 11 Days		Total	
n (observations)	138		122		151		411	
**Study duration**	**n**	**%**	**n**	**%**	**n**	**%**	**n**	**%**
0 h	38	28%	6	5%	145	96%	189	46%
1–2 h	45	33%	11	9%	3	2%	59	14%
3–4 h	41	30%	23	19%	2	1%	66	16%
5–6 h	11	8%	37	30%	0	0%	48	12%
≥7 h	3	2%	45	37%	1	1%	49	12%
**Control variables**	**m**	**sd**	**m**	**sd**	**m**	**sd**	**m**	**sd**
sleep duration (hours)	6.8	1.5	7.4	1.4	7.8	2.1	7.4	1.7
proximity to exam (days^(−1)^)	0.014	0.000	0.323	0.360	0.207	0.129	0.177	0.245
	**n**	**%**	**n**	**%**	**n**	**%**	**n**	**%**
compulsory classes (yes)	138	100%	0	0%	0	0%	138	34%
paid work (yes)	7	5%	9	7%	17	11%	33	8%
	**m**	**sd**	**m**	**sd**	**m**	**sd**	**m**	**sd**
questionnaire completion (hour, h.h)	18.8	1.8	18.2	1.8	18.2	1.8	18.4	1.8

**Table 2 healthcare-11-01674-t002:** Predicting fatigue, vigor, and distress: GEE-Regression coefficients (B), standard error (S.E.), and Wald Chi^2^ statistic (1, n = 411). *p* is provided in the following categories ** *p* < 0.001, * *p* < 0.05.

	Fatigue			Vigor			Distress		
	B	S.E.	χ^2^	B	S.E.	χ^2^	B	S.E.	χ^2^
**Study duration**									
≥7 h	6.3	1.3	23.023 **	−0.98	1.3	0.591	4.8	0.79	36.858 **
5–6 h	3.3	1.2	6.923 *	−1.2	1.2	0.932	4.8	1.0	21.665 **
3–4 h	0.85	1.1	0.599	0.22	1.1	0.040	2.8	0.65	18.367 **
1–2 h	1.2	1.2	0.956	−0.86	1.2	0.520	1.7	0.76	4.89 *
0 h	ref			ref			ref		
**Control variables**									
sleep duration (hours)	−0.85	0.2	13.624 **	0.26	0.24	1.131	0.06	0.11	0.382
yesterday’s compulsory classes (yes)	0.75	1.2	0.408	−0.80	1.0	0.639	−0.79	0.64	1.516
yesterday’s paid work (yes)	2.1	2.2	0.939	0.25	2.1	0.014	1.3	1.5	0.790
proximity to exam (days^(−1)^)	6.8	1.6	17.700 **	−8.8	1.4	39.911	3.3 **	1.2	7.540 **
questionnaire completion (hour)	0.26	0.19	1.824	−0.38	0.2	2.559	0.07	0.16	0.200
gender (male)	1.5	2.3	0.412	−2.3	1.9	1.404	−0.24	1.2	0.040

## Data Availability

Upon request, the corresponding author will provide the data for private use.

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
