# Peer review of "Mentally Demanding Work and Strain: Effects of Study Duration on Fatigue, Vigor, and Distress in Undergraduate Medical Students"

_healthcare, 2023, doi:10.3390/healthcare11121674_

Round 1

Reviewer 1 Report

Overall, the article provides valuable insights into the effects of short durations of studying on medical students' next-day strain reactions, notably fatigue and distress. However, there are some areas where the article could be improved:

  1. The article could benefit from a clearer and more concise introduction that lays out the research question and objectives of the study.
  2. The methods section could be more detailed, particularly with regard to how the data were collected and analyzed. This would improve the transparency and replicability of the study.
  3. The discussion section could be expanded to provide more context for the findings and to discuss potential implications for medical education and student well-being.
  4. The article could benefit from a more thorough review of the existing literature on the topic, including both theoretical and empirical work.
  5. Finally, the article could be improved by providing more specific recommendations for future research, such as exploring the relationship between study duration and specific outcomes or investigating the effectiveness of different recovery strategies for medical students.

in particular:

Here are some possible ways to improve the abstract:

  1. Provide more context: The abstract could start by providing some context on the importance of studying workhours and strain reactions. For example, it could mention that these factors have implications for individual well-being, work productivity, and occupational health and safety.
  2. Clarify the research question: The abstract could be clearer about the specific research question addressed in the study. For example, it could state explicitly that the study aimed to investigate the effects of short durations of mentally demanding work on subjective strain reactions in students.
  3. Summarize the methods: The abstract could provide a brief summary of the study's methods, such as the repeated measure observational design, the assessment of subjective fatigue, vigor, and distress, and the use of Generalized Estimating Equations for data analysis.
  4. Highlight the main findings: The abstract could highlight the main findings of the study more clearly. For example, it could state that short durations of mentally demanding work (i.e., more than 1-2 hours or 5-6 hours) were associated with increased levels of distress or fatigue, respectively, compared to not studying at all.
  5. Discuss the implications: The abstract could briefly discuss the implications of the study's findings. For example, it could suggest that students and educators should be aware of the potential negative effects of short durations of mentally demanding work on well-being and consider strategies to mitigate these effects.

The introduction provides a comprehensive overview of the impact of mentally demanding work on strain reactions, including fatigue and distress, and how this affects the performance, safety, well-being, and health of workers. The author notes that there is general agreement that mentally demanding work leads to strain reactions, and this increases with the duration of work. The author also discusses the types of studies that have been conducted to investigate the impact of mentally demanding work, including experimental studies and real-life observational studies.

The introduction could be improved by providing a more specific research question or objective for the study. It would be helpful to know what specific aspect of the impact of mentally demanding work the study aims to investigate, and how this study adds to the existing body of research. Additionally, the introduction could be made more concise by removing some of the unnecessary background information, such as the details of different types of studies that have been conducted, and focusing on the specific research question or objective.

here are some ways to improve the "Materials & Methods" section:

  1. Provide more detail on data selection:
    • Specify how faulty or missing data was identified and handled.
    • Explain why data entries after 22 o'clock or before 14 o'clock were excluded.
    • Elaborate on the criteria for excluding participants who engaged in employed work of more than 20 hours per week.
  2. Provide more information on study participants:
    • Provide more detail on how participants were recruited and selected.
    • Explain why medical students in their second semester were chosen as the study population.
    • Provide more information on the demographics of the participants (e.g. ethnicity, socioeconomic status, etc.).
  3. Clarify the study design:
    • Provide more information on how the study was structured and how the data was collected.
    • Explain why the study was conducted over 21 days and why there were three different phases.
    • Clarify the timing of the study in relation to the academic semester.
  4. Provide more detail on variables:
    • Provide more information on how the independent variable "study-duration" was measured and why it was recoded into categories.
    • Clarify how the control variables were measured and why they were included in the study.
    • Explain why the dependent variables (fatigue, vigor, and distress) were chosen and how they were assessed.
    •  
  5. Provide references:
    • If the study design or methods were based on previous research, it can be helpful to provide references to those sources.

Some possible ways to improve the  results:

  1. Providing more context: While the table provides some summary statistics, it may be helpful to provide more background information on why certain variables were measured and how they relate to the research questions.
  2. Clarifying the results: While the results are summarized in Table 2 and illustrated in Figure 1, it may be helpful to provide more details on the statistical tests used and the exact values obtained.
  3. Using clearer language: The language used in this section is somewhat technical and may be difficult for some readers to understand. Using simpler, more straightforward language can help make the results more accessible.

Here are some suggestions to improve the results section:

  1. Provide more details about the participants: It would be helpful to include some basic demographic information about the participants, such as age, gender, and academic level. This can help readers understand the generalizability of the study's findings.
  2. Include effect sizes: It would be beneficial to include effect sizes in addition to statistical significance testing. This information can help readers understand the practical significance of the study's findings.
  3. Discuss limitations: It's important to discuss limitations of the study that may affect the interpretation of the results. For example, the study only investigated the association between study duration and psychological strain in students, so it may not be generalizable to other populations.
  4. Compare findings to previous research: It would be helpful to discuss how the findings of this study compare to previous research on the same topic. This can help contextualize the results and highlight any novel findings.

Here are some suggestions to improve the Conclusions section:

  1. Provide a brief summary of the main findings of the study: Start the conclusion by providing a brief summary of the main findings of the study. This will help readers to quickly understand the key takeaways from your research.
  2. Emphasize the significance of the findings: Explain why the findings of the study are important and how they contribute to the existing knowledge in the field. Highlight the significance of the study in terms of its implications for students and academic institutions.
  3. Acknowledge limitations of the study: It is important to acknowledge the limitations of the study and how they may affect the interpretation of the results. This shows that you have considered the weaknesses of your research and provides an opportunity for future studies to address these limitations.
  4. Provide recommendations for future research: Based on the limitations of the study, provide recommendations for future research that could help to address these limitations and further advance knowledge in the field.
  5. Consider the implications for practice: In addition to implications for research, consider the practical implications of the findings for students and academic institutions. Provide recommendations for how these implications could be addressed in practice.

The English used in the article appears to be written at a high level. The sentences are well-constructed and the vocabulary used is appropriate for an academic paper. However, there may be room for improvement in terms of clarity and conciseness, as some sentences are quite long and complex, which could make them difficult to understand for some readers. Additionally, there are a few minor grammatical errors and typos throughout the paper that could be corrected. Overall, with some editing and revision, the English used in the article could be further improved.

Author Response

Dear Reviewers,

Thank you for reading our manuscript “Mentally Demanding Work and Strain: Effects of Study Duration on Fatigue, Vigor and Distress in Students” and for the opportunity to submit a revised version. Thank you for your detailed comments and specific suggestion on how to improve our presentation. They challenged us to advance our manuscript.

We are confident that we were able to address all issues suggested and to improve the manuscript. Due to the complexity of the changes, many parts of the text had to be restructured and rewritten, thus changes could no longer be tracked. We provide a point-by-point list elaborating on the revisions made, and when illustrative, we provide highlights in different color in the text. 

We are looking forward to the next steps and are prepared to work efficiently.

Kind regards,

Gerhard Blasche

Comments and Suggestions for Authors (Combined Reviewer 1 and Reviewer 2)

Reviewer 1: Overall, the article provides valuable insights into the effects of short durations of studying on medical students' next-day strain reactions, notably fatigue and distress. However, there are some areas where the article could be improved

By focusing on the particular suggestions, we were able to improve the five issues listed below. We comment what we have done at each of the specific issues.

  1. The article could benefit from a clearer and more concise introduction that lays out the research question and objectives of the study.

  1. The methods section could be more detailed, particularly with regard to how the data were collected and analyzed. This would improve the transparency and replicability of the study.

  1. The discussion section could be expanded to provide more context for the findings and to discuss potential implications for medical education and student well-being.

  1. The article could benefit from a more thorough review of the existing literature on the topic, including both theoretical and empirical work.

  1. Finally, the article could be improved by providing more specific recommendations for future research, such as exploring the relationship between study duration and specific outcomes or investigating the effectiveness of different recovery strategies for medical students.

Reviewer 2: (introstatement) Thank you for giving me this opportunity to review this manuscript, “Mentally Demanding Work and Strain: Effects of Study Duration on Fatigue, Vigor and Distress in Students”. This study aims to explore the influence of study duration, as an indicator of mentally challenging work in students, on subsequent day's stress responses, using a repeated measures observational design. The primary advantages of this study stem from its utilization of a longitudinal research design and its grounding in real-life situations. The employed methods are notably impressive, and the findings yield considerable intrigue. I have only some minor concerns about the manuscript which are outlined below and I sincerely hope that the authors find them helpful in any future revisions of their work.

Title:

Reviewer 2:

1. Please consider adding information about the research subjects in the title, such as "medical school students".

The new title reads nicely with “medical students” and goes along with the improved abstract as suggested by Rev1. (green highlights).

Reviewer 1:

in particular:

Here are some possible ways to improve the abstract:

  1. Provide more context: The abstract could start by providing some context on the importance of studying workhours and strain reactions. For example, it could mention that these factors have implications for individual well-being, work productivity, and occupational health and safety.
  2. Clarify the research question: The abstract could be clearer about the specific research question addressed in the study. For example, it could state explicitly that the study aimed to investigate the effects of short durations of mentally demanding work on subjective strain reactions in students.
  3. Summarize the methods: The abstract could provide a brief summary of the study's methods, such as the repeated measure observational design, the assessment of subjective fatigue, vigor, and distress, and the use of Generalized Estimating Equations for data analysis.
  4. Highlight the main findings: The abstract could highlight the main findings of the study more clearly. For example, it could state that short durations of mentally demanding work (i.e., more than 1-2 hours or 5-6 hours) were associated with increased levels of distress or fatigue, respectively, compared to not studying at all.
  5. Discuss the implications: The abstract could briefly discuss the implications of the study's findings. For example, it could suggest that students and educators should be aware of the potential negative effects of short durations of mentally demanding work on well-being and consider strategies to mitigate these effects.

Thank you for suggesting feasible ways for improving the abstract. We were able to make all the modifications as suggested. And we structured the Abstract using Headings Methods-Results-Conclusions

INTRODUCTION

Reviewer 1:

The introduction provides a comprehensive overview of the impact of mentally demanding work on strain reactions, including fatigue and distress, and how this affects the performance, safety, well-being, and health of workers. The author notes that there is general agreement that mentally demanding work leads to strain reactions, and this increases with the duration of work. The author also discusses the types of studies that have been conducted to investigate the impact of mentally demanding work, including experimental studies and real-life observational studies.

The introduction could be improved

… by providing a more specific research question or objective for the study.

We specified the research question (lines 82-84; pink).

Also there are hypotheses specified (lines 99-104; without highlight).

… and how this study adds to the existing body of research

…. And added information on how the study adds: lines 90-92

… It would be helpful to know what specific aspect of the impact of mentally demanding work the study aims to investigate

We specify the aspect of impact in the paragraph (lines 93-99; green highlights). 

Additionally, the introduction could be made more concise by removing some of the unnecessary background information, such as the details of different types of studies that have been conducted, and focusing on the specific research question or objective.

The part of the text elaboration on the details of different types of studies have been restructured to only give the information needed to support research questions. Other details on different types of studies have been removed (without highlights).

METHOD

Reviewer 2:

2. Line 121: 2.1 Data Selection. This subsection introduces the process of obtaining the final dataset, particularly the reasons for participant exclusion. It is advised to reposition this subsection after the introduction of participant recruitment. This sequence appears more logical, as it first explains the method and number of recruited participants before discussing the exclusion criteria.

We took up the suggestion to work on the sequence of the subchapters.

To support reading, the subchapters in the Method section where rearranged and now appear in this order:

2.1 Study participants

2.2 Design

2.3Variables

2.4 Data Analysis

2.5 Data Selection

(yellow highlights in the section headings)

Reviewer: 1

Here are some ways to improve the "Materials & Methods" section:

Provide more detail on data selection:

·       Specify how faulty or missing data was identified and handled

We added this information in lines 146-147 (yellow highlights).

  • Explain why data entries after 22 o'clock or before 14 o'clock were excluded.

Line 215-217, (yellow highlights).

·       Elaborate on the criteria for excluding participants who engaged in employed work of more than 20 hours per week.

Line 108-109 specifies the program as full-time program, and line 210-2013 justifies exclusion of >20hrs as being against recommendations.

1        Provide more information on study participants:

·        Provide more detail on how participants were recruited and selected.

Lines 118-119 (green highlights).

·        Explain why medical students in their second semester were chosen as the study population.

Lines 111-114 (grey highlights).

·        Provide more information on the demographics of the participants (e.g. ethnicity, socioeconomic status, etc.).

We provided the available information on the demographics of the population in lines 107-111 (blue text).

Reviewer 2: (Design section)

3. Line 149: Regarding how the observations in your article were obtained, can you provide examples to illustrate?

We rewrote the Design section (2.2. Design) entirely when responding to comments made by Rev 1 and Rev 2 – and included the information how we obtained the information (lines 125-147; without highlights).

Reviewer 2:

4.Lines 147-159: The study procedure, consisting of three phases, is introduced here. Consider utilizing a table or flowchart to present this information, as it will enhance readability for the reader.

We entered information in the section “Study Design” to clarify the three phases (lines 126-129; blue text).

Reviewer 1:

Clarify the study design:

    • Provide more information on how the study was structured and how the data was collected.
    • Explain why the study was conducted over 21 days and why there were three different phases.
    • Clarify the timing of the study in relation to the academic semester

Reviewer 2

5.Lines 165-167: The text states that "55% of the 7 or more hour category had studied for 7-8 hours, … more than 12 hours." This information represents the study's results and seems not be placed under the "variables" subsection.

We removed the respective information in this section (no highlights).

Moreover, since this study is longitudinal, it is unclear if the reported figures are averages, obtained at a specific time, or something else. Please provide clarification on this matter

Where necessary, we specified what type of information is in a figure (line 250-251; yellow highlights).

Reviewer 2:

6. Lines 168-170:  It is unclear why the "time of questionnaire completion" is treated as a control variable. Please provide a rationale for this decision.

We elaborated on why each of the control variables was included (line 159/160 pink highlights).

1       Provide more detail on variables:

Provide more information on how the independent variable "study-duration" was measured and why it was recoded into categories

Lines 152-155; olive green.

2       Clarify how the control variables were measured and why they were included in the study.

We elaborated on the control Variable as follows in the lines 156-166; blue text.

·        Explain why the dependent variables (fatigue, vigor, and distress) were chosen and how they were assessed.

We elaborated on the dependent variables (167-176; green text).

Reviewer 2:
7. 2.5 Data Analysis:

(1) The data analysis followed the Generalized Estimating Equations (GEE) procedure. To facilitate better understanding, please provide more information on GEE, including its definition and the reasons for choosing GEE over alternative methods.

We now explain what GEE is and explain why we used it four our study (line 188-191; red highlights).

(2) Ensure that the data analysis section corresponds with the results section. For instance, the method for obtaining the p-value in Table 1 is not explained.

We solved this problem by elaborating how we use the GEE in the Method section (lines 195-197; violet highlights).

3       Provide references:

·        If the study design or methods were based on previous research, it can be helpful to provide references to those sources.

We provided two references showing the application of GEE Models for clustered data (light blue highlights in line 189-191).

RESULTS
Some possible ways to improve the  results:

1.     Providing more context: While the table provides some summary statistics, it may be helpful to provide more background information on why certain variables were measured and how they relate to the research questions.

In the respective section of the results we elaborated on the variables and provided more details in text and subheadings (line 227 & line 240).

Reviewer 2:

8. Line 219: The χ2 value is not provided in Table 2. Please considered including this information.

We added exact values of standard errors and Wald Chi2 test statistics in text (lines 231, 234, 235, 238, and in table, 2 line 274).

Reviewer 1:

2.     Clarifying the results: While the results are summarized in Table 2 and illustrated in Figure 1, it may be helpful to provide more details on the statistical tests used and the exact values obtained.

3.     Using clearer language: The language used in this section is somewhat technical and may be difficult for some readers to understand. Using simpler, more straightforward language can help make the results more accessible.

We rewrote the whole section, without highlights using shorter sentences and more common words.

Here are some suggestions to improve the results section:

1.     Provide more details about the participants: It would be helpful to include some basic demographic information about the participants, such as age, gender, and academic level. This can help readers understand the generalizability of the study's findings.

We included information about age and gender in the method section 2.1. “Study participants” (line 120-124; pink highlights).

4.     Include effect sizes: It would be beneficial to include effect sizes in addition to statistical significance testing. This information can help readers understand the practical significance of the study's findings.

Thank you for emphasizing the benefit of effect size in reporting results. However, to account for the clustering in the data due to the longitudinal, repeated measurement design, we used GEE. Unfortunately, there is currently no effect size measure included in the GENLIN Procedure of SPSS. This procedure adjusts for correlated data to receive more accurate values for significance testing, which we believe is more important than providing effect sizes (line 197f ; grey highlights).

5.     Discuss limitations: It's important to discuss limitations of the study that may affect the interpretation of the results. For example, the study only investigated the association between study duration and psychological strain in students, so it may not be generalizable to other populations.

We discussed limitations (see limitation section).

1.     Compare findings to previous research: It would be helpful to discuss how the findings of this study compare to previous research on the same topic. This can help contextualize the results and highlight any novel findings.

CONCLUSION

Here are some suggestions to improve the Conclusions section:

  1. Provide a brief summary of the main findings of the study: Start the conclusion by providing a brief summary of the main findings of the study. This will help readers to quickly understand the key takeaways from your research.

We included the recommend brief summary of main results (lines 291-297; green highlights).

Conclusion: Line 391-396; yellow highlights.

  1. Emphasize the significance of the findings: Explain why the findings of the study are important and how they contribute to the existing knowledge in the field. Highlight the significance of the study in terms of its implications for students and academic institutions.

We elaborate on the significance of the findings in the conclusion (line 397 ff. and 403 ff.; pink highlights).

  1. Acknowledge limitations of the study: It is important to acknowledge the limitations of the study and how they may affect the interpretation of the results. This shows that you have considered the weaknesses of your research and provides an opportunity for future studies to address these limitations.

We build a new subchapter 4.1. Limitations (line 366; yellow highlights).

  1. Provide recommendations for future research: Based on the limitations of the study, provide recommendations for future research that could help to address these limitations and further advance knowledge in the field.

We included suggestions for future research in line 388 (blue highlights) and in line 413 in the conclusion (blue highlights).

  1. Consider the implications for practice: In addition to implications for research, consider the practical implications of the findings for students and academic institutions. Provide recommendations for how these implications could be addressed in practice.

We reflect on recommendation on implications (line 407 ff.; blue text) in the conclusion. 

Comments on the Quality of English Language

The English used in the article appears to be written at a high level. The sentences are well-constructed and the vocabulary used is appropriate for an academic paper. However, there may be room for improvement in terms of clarity and conciseness, as some sentences are quite long and complex, which could make them difficult to understand for some readers.

We checked all sentences and broke down sentences running over more than three lines, thus also working on improvements in terms of clarity (no highlights possible due to the high number of changes).

Additionally, there are a few minor grammatical errors and typos throughout the paper that could be corrected. Overall, with some editing and revision, the English used in the article could be further improved

We made all effort possible for us to further reduce the number of grammatical errors and typos.

Reviewer 2 Report

Thank you for giving me this opportunity to review this manuscript, “Mentally Demanding Work and Strain: Effects of Study Duration on Fatigue, Vigor and Distress in Students”. This study aims to explore the influence of study duration, as an indicator of mentally challenging work in students, on subsequent day's stress responses, using a repeated measures observational design. The primary advantages of this study stem from its utilization of a longitudinal research design and its grounding in real-life situations. The employed methods are notably impressive, and the findings yield considerable intrigue. I have only some minor concerns about the manuscript which are outlined below and I sincerely hope that the authors find them helpful in any future revisions of their work.

1. Please consider adding information about the research subjects in the title, such as "medical school students".

2. Line 121: 2.1 Data Selection. This subsection introduces the process of obtaining the final dataset, particularly the reasons for participant exclusion. It is advised to reposition this subsection after the introduction of participant recruitment. This sequence appears more logical, as it first explains the method and number of recruited participants before discussing the exclusion criteria.

3. Line 149: Regarding how the observations in your article were obtained, can you provide examples to illustrate?

4.Lines 147-159: The study procedure, consisting of three phases, is introduced here. Consider utilizing a table or flowchart to present this information, as it will enhance readability for the reader.

5.Lines 165-167: The text states that "55% of the 7 or more hour category had studied for 7-8 hours, … more than 12 hours." This information represents the study's results and seems not be placed under the "variables" subsection. Moreover, since this study is longitudinal, it is unclear if the reported figures are averages, obtained at a specific time, or something else. Please provide clarification on this matter.

6. Lines 168-170:  It is unclear why the "time of questionnaire completion" is treated as a control variable. Please provide a rationale for this decision.

7. 2.5 Data Analysis: (1) The data analysis followed the Generalized Estimating Equations (GEE) procedure. To facilitate better understanding, please provide more information on GEE, including its definition and the reasons for choosing GEE over alternative methods. (2) Ensure that the data analysis section corresponds with the results section. For instance, the method for obtaining the p-value in Table 1 is not explained.

8. Line 219: The χ2 value is not provided in Table 2. Please considered including this information.

Author Response

(The authors gave the same response as above.)

Round 2

Reviewer 1 Report

  1. Provide more clarity: Certain sections of the manuscript, such as the Materials & Methods and Results sections, contain long and complex sentences. Consider breaking them down into shorter, more concise sentences to improve readability and comprehension.
  2. Enhance organization: The manuscript could benefit from improved organization and structure. Consider using subheadings or bullet points to present information in a more organized and easily digestible manner. This will help readers navigate the content more effectively.
  3. Expand on background information: Provide a more comprehensive introduction that includes a clear overview of the topic, a review of relevant literature, and a rationale for the study. This will provide readers with a better understanding of the context and importance of the research.
  4. Address limitations and future directions: It is important to explicitly discuss the limitations of the study and acknowledge any potential biases or constraints. Additionally, provide suggestions for future research directions or potential areas of improvement to enhance the study's findings and contribute to the field.
  5. Improve the conclusion: As mentioned earlier, the conclusion could be strengthened by providing a clear and concise summary of the main findings and their implications. Ensure that the conclusion effectively highlights the study's key takeaways and offers actionable insights for researchers or practitioners in the field.
  1. Proofread and edit: Review the manuscript for grammatical errors, typos, and inconsistencies. A thorough proofreading and editing process will help improve the overall clarity and professionalism of the manuscript.

examples:

  1. "On a theoretical level, our study elaborates the complex relation between effort to perform": Here, it seems that the verb "elaborates" should agree with the subject "our study" in the third person singular form, so it should be "elaborates" instead of "elaborate."
  2. "Increasingly and repeatedly straining part of one’s physiological and psychological resources": The phrase "straining part" lacks a clear verb and may benefit from rephrasing. For example, it could be revised to "Repeatedly straining a portion of one's physiological and psychological resources."
  3. "students were able to maintain a rather constant level vigour": The phrase "a rather constant level vigour" is missing the article "of" before "vigour." It should be "a rather constant level of vigor."
  4. "Taking into account the time untill the deadline": The word "untill" should be corrected to "until" to properly convey the meaning of the sentence.
  5. "For full recovery some days after the deadline are necessary": The verb "are" should agree with the singular subject "full recovery." It should be "For full recovery, some days after the deadline is necessary."

Author Response

Comments and Suggestions for Authors

  1. Provide more clarity: Certain sections of the manuscript, such as the Materials & Methods and Results sections, contain long and complex sentences. Consider breaking them down into shorter, more concise sentences to improve readability and comprehension.

  • We revised the text throughout the manuscript, among others splitting long sentences in the Material and Method Section, also with the help of an AI spell-checker. All changes are made visible using track changes.

  1. Enhance organization: The manuscript could benefit from improved organization and structure. Consider using subheadings or bullet points to present information in a more organized and easily digestible manner. This will help readers navigate the content more effectively.

  • Thank you for this suggestion. We introduced several subheadings to improve structure and readability in the method section and result sections. In addition, we incerted bullet points to improve readability. All changes are made visible using track changes.

  1. Expand on background information: Provide a more comprehensive introduction that includes a clear overview of the topic, a review of relevant literature, and a rationale for the study. This will provide readers with a better understanding of the context and importance of the research.

  • We expanded background information in the introduction section (e.g. third paragraph) to provide readers with a better understanding of the context and importance of the research. All changes are made visible using track changes.

  1. Address limitations and future directions: It is important to explicitly discuss the limitations of the study and acknowledge any potential biases or constraints.

  • We addressed limitations (section 4.1. “Limitations”), thereby addressing potential biases and constraints of our study.

  1. Additionally, provide suggestions for future research directions or potential areas of improvement to enhance the study's findings and contribute to the field.

  • We provided suggestions for future research (5.0. “Conclusions”, last paragraph) based on the results and limitations of our own study.

  1. Improve the conclusion: As mentioned earlier, the conclusion could be strengthened by providing a clear and concise summary of the main findings and their implications. Ensure that the conclusion effectively highlights the study's key takeaways and offers actionable insights for researchers or practitioners in the field.

  • We strived to improve the conclusions by rephrasing the summary of the main findings and elaborating on theoretical and practical implications.

Comments on the Quality of English Language

  1. Proofread and edit: Review the manuscript for grammatical errors, typos, and inconsistencies. A thorough proofreading and editing process will help improve the overall clarity and professionalism of the manuscript.

examples:

  1. "On a theoretical level, our study elaborates the complex relation between effort to perform": Here, it seems that the verb "elaborates" should agree with the subject "our study" in the third person singular form, so it should be "elaborates" instead of "elaborate."
  2. "Increasingly and repeatedly straining part of one’s physiological and psychological resources": The phrase "straining part" lacks a clear verb and may benefit from rephrasing. For example, it could be revised to "Repeatedly straining a portion of one's physiological and psychological resources."
  3. "students were able to maintain a rather constant level vigour": The phrase "a rather constant level vigour" is missing the article "of" before "vigour." It should be "a rather constant level of vigor."
  4. "Taking into account the time untill the deadline": The word "untill" should be corrected to "until" to properly convey the meaning of the sentence.
  5. "For full recovery some days after the deadline are necessary": The verb "are" should agree with the singular subject "full recovery." It should be "For full recovery, some days after the deadline is necessary."

  • We proofread the manuscript thoroughly and sought to correct any phrasing and/or grammar errors.